# Magnetotactic bacteria in a droplet self-assemble into a rotary motor

Benoit Vincenti [1], Gabriel Ramos [2], Maria Luisa Cordero [2], Carine Douarche [3], Rodrigo Soto [2] & Eric Clement [1]*

From intracellular protein trafficking to large-scale motion of animal groups, the physical concepts driving the self-organization of living systems are still largely unraveled. Self-organization of active entities, leading to novel phases and emergent macroscopic properties, recently shed new light on these complex dynamical processes. Here we show that under the application of a constant magnetic field, motile magnetotactic bacteria confined in water-in-oil droplets self-assemble into a rotary motor exerting a torque on the external oil phase. A collective motion in the form of a large-scale vortex, reversable by inverting the field direction, builds up in the droplet with a vorticity perpendicular to the magnetic field. We study this collective organization at different concentrations, magnetic fields and droplet radii and reveal the formation of two torque-generating areas close to the droplet interface. We characterize quantitatively the mechanical energy extractable from this new biological and self-assembled motor.

[1] Laboratoire PMMH, UMR 7636 CNRS-ESPCI-Sorbonne Université-Université Paris Diderot, 7-9 quai Saint-Bernard, 75005 Paris, France. [2] Departamento de Física, FCFM, Universidad de Chile, Av. Blanco Encalada 2008, Santiago, Chile. [3] Laboratoire FAST, UMR 7608, Univ. Paris-Sud, CNRS, Université Paris-Saclay, F-91405 Orsay, France. *email: eric.clement@upmc.fr

One of the hallmark of life in its most phenomenological appearance is its ability to spontaneously produce collective, hierarchical, and organized motion. However, it has been shown that non-living entities can also develop complex and coordinated structures when driven out of thermodynamic equilibrium[1]. So far, a complete picture guiding the emergence of these active patterns is still lacking. Indeed, since the pioneering work of Vicseck[2], physics communities have constantly tried to unravel the basic principles leading to collective pattern formation of active particles[3,4]. Interestingly, this quest has inspired the design of new materials and devices[5–10].

Like the many artificial active systems recently proposed to tackle this question[11–15], assemblies of motile bacteria turned out to be a rich and insightful experimental playground[16–23]. Among the rich topics that were investigated, the confinement of bacteria and of active particles has been the focus of many experimental[24–27] and theoretical studies[27,28], showing that, under strong confinement, vortical collective motions may spontaneously appear.

A class of bacteria—called magnetotactic (MTB)—can grow internally a microscopic magnet, hence providing an external handle to drive their swimming orientation[29,30]. As a source of nano-magnetic particles widely used in a medical context, MTB are micro-organisms of strong practical interest[31]. For example, the magnetic alignment, combined with a micro-aerotactic swimming response, qualifies such micro-swimmers as a promising vector for targeted drug therapy[32]. Recently, it was proposed, on theoretical grounds, that a suspension of such MTB could display original magneto-rheological properties[33,34], novel pattern formation[35] and hydrodynamic instabilities[36,37]. In particular, the pearling hydrodynamic instability reported by Waisbord et al.[38], the velocity condensation[39] and the emergence of new phases induced by a magnetic field[40] are striking examples of these.

Here, we study aqueous spherical droplets suspended in oil and containing a suspension of MTB. We show how MTB self-assemble into a rotary motor under the application of a uniform and constant magnetic field, providing a mechanical torque to the fluid outside the droplets. In the self-assembly process, the magnetic field induces an accumulation of the swimming bacteria in diametrically opposed areas at the surface of the droplet. At high bacterial concentration, the flows resulting from the swimming activity of the bacteria and originating from these areas, interact to create a collective solid-like vortex flow in the central droplet core. Through Particle Image Velocimetry (PIV) analysis and particle tracking, we quantify the flows inside and outside the droplet and measure the net torque produced by this micromotor as a function of the magnetic field and the droplet radius. Finally, we provide an explanation on how an external torque can be generated despite the fact that the swimmers self-propel at almost zero Reynolds number.

## Results

**Experimental setup.** A water-in-oil emulsion is prepared by shaking a mixture of hexadecane oil with a suspension of MTB (*Magnetospirillum gryphiswaldense* MSR-1) (see Fig. 1a and Supplementary Movie 1). With our preparation protocol for bacteria[41] (see "Methods" and Supplementary Note 3 for details), MTB swim with a velocity $V_0$ ranging from 20 to 40 μm s$^{-1}$ and exhibit a magnetic moment $m \sim 10^{-16}$ J T$^{-1}$. The droplets encapsulate an almost even population of *north-seeker* (NS) and *south-seeker* (SS) bacteria, meaning that, under the application of a magnetic field, roughly half of the population will swim persistently towards the (magnetic) north and the other half towards the south. The emulsion is placed between two glass slides on the

stage of an inverted microscope and at the center of a pair of Helmholtz coils, where a constant horizontal magnetic field is generated. The droplet radius, $R$, spans typically from 20 to 120 μm. Once the emulsion is formed, all the bacteria dwell in the aqueous phase. We call north pole (NP) the point on the droplet surface corresponding to the far-most position in the direction of the magnetic north, and south pole (SP) the diametrically opposed position (see Fig. 1c).

**Vortex flow inside the droplets.** In absence of magnetic field, regardless of the bacteria density, the swimming direction of MTB in the drops is random and unbiased. When the suspension is dense no collective motion is observed at the droplet larger scale; only fluctuating and intermittent vortices appear, with typical sizes much smaller than the droplet diameter. Note that the experiments of Wioland et al.[24] show the development of global vortices taking place at higher volume fraction.

When a magnetic field is set and in dilute conditions (for a cell density $n \sim 10^{14}$ bact m$^{-3}$), NS (resp. SS) bacteria accumulate in the vicinity of the north (resp. south) poles of the droplet as a consequence of the bacteria swimming persistence described above (see Fig. 2a). Because NS and SS bacteria are performing reversals, we observe some bacteria escaping from the accumulation regions (see Supplementary Movie 2). At an intermediate density (typically $n \sim 10^{15}$ bact m$^{-3}$), the accumulation pattern becomes more unstable with episodic formation of jets propelling the fluid and the bacteria out of the polar positions, thus creating two local recirculation zones near each pole (see Supplementary Movie 3 and Fig. 2b). For a dense suspension ($n \sim 10^{17}$ bact m$^{-3}$), which is the case in the rest of the study, a steady and uniform collective rotational motion is observed, with an axis of rotation perpendicular to the magnetic field (see Supplementary Movie 4 and Fig. 2 (c)) and oriented along the gravity direction. Although all symmetric planes containing the magnetic field direction could have been chosen by the bacteria, the vortex is actually rotating in the $x$-$y$ plane. This might be due to a sedimentation process (about 20% denser than the medium[42]) which yields a stable stratified suspension. Visualization in the other horizontal planes shows a similar rotation field of equal direction as in the equatorial plane (see Supplementary Movies 5 and 6). The rotation direction chosen by the fluid is not completely random, with approximately 84% of the drops rotating in a clockwise (CW) direction looked from the top. This preferential choice of spontaneous rotation is not completely elucidated yet but may be related to the helicity of MTB. An interesting property of this collective rotational motion is that, regardless of the choice of rotation direction at the magnetic field onset, the direction can be reversed by reversing the magnetic field (see Supplementary Movie 7). Some experiments were performed with a larger fraction of NS bacteria, selecting them using a macroscopic magnet. At the scale of the droplets, some SS individuals are recovered, but in significantly less quantity than the NS. Even with such an unbalanced ratio of NS/SS, we found that the collective rotation is still preferentially CW.

From now on, we focus on the characteristics of the vortex flow at a fixed density $n \sim 10^{17}$ bact m$^{-3}$. For a magnetic field larger than a threshold value (typically $0.4 \pm 0.1$ mT), one observes the emergence of the large scale vortical flow previously mentioned. At the experimental density, individual bacteria are not observable, however, through PIV analysis of phase-contrast microscopy images, we obtain the temporally and spatially resolved velocity fields due to the bacterial motion inside the drops $\mathbf{V}^d(x, y)$ (Fig. 2d–f). The flow geometry is reminiscent of the experiments by Sokolov et al.[43], but here the bacteria are aligned along the magnetic field. The flow field shows a central

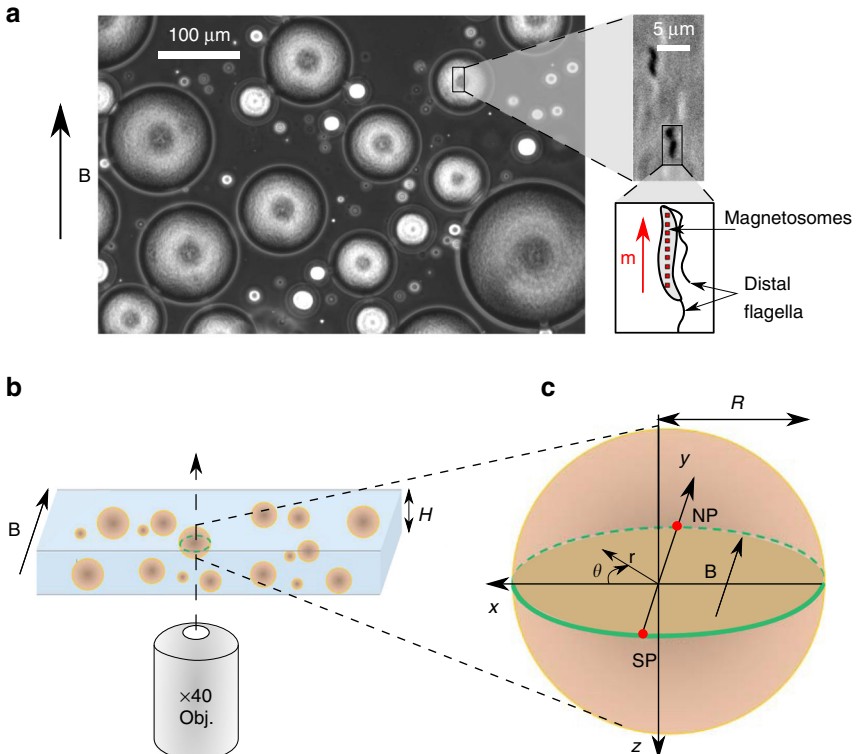

**Fig. 1** Water-in-oil emulsion of magnetotactic bacteria. **a** A ×10 phase-contrast image of an emulsion of magnetotactic bacteria (bacteria remain inside droplets) in hexadecane oil. A magnetic field of 4 mT is applied as indicated by the arrow (see the corresponding Supplementary Movie 1). A broad distribution of droplets radii is obtained, spanning typically from 20 to 120 μm. Zoom in: ×40 phase-contrast image of two magnetotactic bacteria *Magnetospirillum gryphiswaldense* MSR-1 (darkest zones) swimming along the magnetic field direction. Zoom in: Sketch of a magnetotactic bacterium carrying magnetosomes (red squares) and two distal flagella. The magnetosomes are aligned along the body, generating a magnetic moment *m*. **b** Setup principle: a droplet, lying on the bottom plate of a pool of height $H = 270$ μm and placed on the stage of an inverted microscope, is observed at its equatorial plane with a ×40 objective. A uniform magnetic field is applied in the observation plane, parallel to the bottom and top plates of the pool. **c** Definitions of the north pole (NP), the south pole (SP) of the droplet and the spatial coordinates. *R* is the droplet radius

vortical structure and presents two maximal streams located near the poles reminiscent of the two jets visualized at lower bacterial concentration. The strength of the vortical flow field increases with the intensity of the magnetic field (Fig. 2e–f). Computation of the angular average of the orthoradial velocity, $\overline{V_\theta^d}(r) = \frac{1}{2\pi}\int_0^{2\pi} V_\theta^d(r,\theta)\,d\theta$, brings evidence for an effective solid-core rotating motion, characterized by an angular velocity $\Omega^d$ ($\overline{V_\theta^d}(r) = \Omega^d r$, see Fig. 3a). The solid-core spans one-half of the droplet radius for all the radii investigated. At increasing magnetic field intensities, the magnitude of $\Omega^d$ increases to saturate at larger magnetic fields. Beyond the droplet core, the suspension is sheared and the velocity decreases down to a non-zero value at the droplet interface. PIV analysis also shows local recirculating regions in a direction opposite to the core rotation (see blue regions enhanced in velocity maps on Fig. 2e–f), shifted with respect to the poles in the direction of the rotating motion.

**Flow in the oily phase.** By tracking 1 μm-diameter melamine resin beads in the surrounding hexadecane oil, we observe a net circular flow outside the droplets (see Supplementary Movie 4 and Fig. 3b), indicating the outcome of a net torque on the fluid outside the droplet. Indeed, although zero-torque sources can generate finite circulations, it can be proven that the circulation must change sign when measured at different planes (see Supplementary Note 4). As shown in Supplementary Movies 4, 5, and 6, the circulation sign is the same for all planes, which can only be

produced by torque sources. Also, zero-torque sources would produce mainly radially oriented flows (Eq. 4 of the Supplementary Note 4), contrary to the orthoradial flows we measure. Hence, the MTB self-assemble inside the droplet to form a rotary motor. The angular average of the orthoradial velocity of the tracers $\overline{V_\theta^{oil}}(r)$ is determined for different outer radii $r$ and for various magnetic field $B$ and droplet radii $R$ (see methods for details). In all cases, we measured a net fluid rotation in the same direction as the central core rotation. However, we observe that local recirculation patterns, opposed to the net fluid rotation, appear close to the poles, mirroring the previously mentioned counter-flow inside the droplet (see Supplementary Movie 4).

**Torque measurements.** In the following, we measure the energy production associated with the rotary motor (i.e., the effective torque acting on the oil) and identify the mechanism of the torque generation inside the droplets. The effective torque exerted by the rotary motor is extracted by fitting the radial dependence of the mean orthoradial velocity in the oil phase with a simple hydrodynamic model (see methods and Supplementary Note 1), consisting in a sphere driven in rotation by a torque $\tau$. However, the drop being sedimented at the bottom of the chamber, a hydrodynamic image of the rotating droplet is added to account for the no-slip boundary condition of the flow field at the solid interface. The dependence of the orthoradial projection of the external flow, $\overline{V_\theta^{oil}} = \frac{1}{2\pi}\int_0^{2\pi} V_\theta^{oil}(r,\theta)\,d\theta$, with the distance $r$ to the

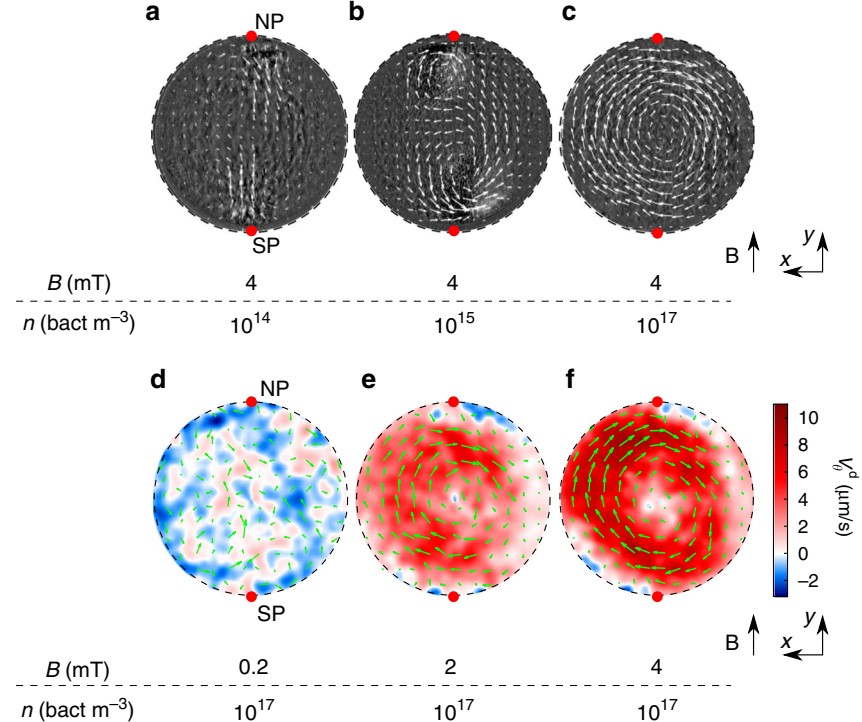

**Fig. 2** Influence of the cell density $n$ and the magnetic field $B$ on the emergence of collective vortical motion. **a–c** ×40 phase-contrast images of droplets superimposed with time average PIV velocity fields (green arrows). We show the influence of the cell density $n$ on the phenomenology, the magnetic field is fixed $B = 4$ mT. (**a**) $n \sim 10^{14}$ bact m$^{-3}$, $R = 43$ μm: Bacteria accumulate at the poles of the droplet. (**b**) $n \sim 10^{15}$ bact m$^{-3}$, $R = 89$ μm: unstable recirculation flows appear at the poles of the droplet. (**c**) $n \sim 10^{17}$ bact m$^{-3}$, $R = 55$ μm: the bacteria self-organize to form a stable vortex flow at the center of the droplet. **d–f** Colored maps of the orthoradial projection of the instantaneous PIV velocity fields $V_\theta^d$ (red-blue colormap, enhancing positive and negative values) superimposed with the instantaneous PIV velocity field (green arrows). The radius of the droplet is constant $R = 83$ μm. We show the influence of the magnetic field magnitude on the phenomenology, the cell density is fixed $n = 10^{17}$ bact m$^{-3}$. **d** $B = 0.2$ mT: no large scale collective motion is observed. **e** $B = 2$ mT: vortex flow centered at the droplet center. **f** $B = 4$ mT: the vortex flow is stronger than at $B = 2$ mT. **e, f** Recirculation flows (negative values of $V_\theta^d$) close to the poles are identified in blue

center of the droplet reads (for $r > R$):

$$\overline{V_\theta^{\text{oil}}}(r) = \frac{\tau}{8\pi\eta_{\text{oil}}R^2}\left[\frac{R^2}{r^2} - \frac{r/R}{\left(r^2/R^2 + 4\right)^{3/2}}\right], \quad (1)$$

where $\eta_{\text{oil}} = 3 \times 10^{-3}$ Pa.s is the dynamic viscosity of hexadecane at 25°C. The dependency with $\frac{R^2}{r^2}$ results from the rotation of a sphere of radius $R$ in the bulk and $-\frac{r/R}{\left(r^2/R^2+4\right)^{3/2}}$ is a correction term accounting for the presence of the bottom plate of the pool. From the experimental measurement of $\overline{V_\theta^{\text{oil}}}(r)$, we are then able to estimate $\tau$ for various droplets radii at different magnetic field intensities. Fig. 3 c shows the dependency of $\tau$ with respect to the core solid rotation of the MTB suspension $\Omega^d$ for different mean radii (each data point corresponds to an average over several droplets of similar radii): we observe that $\tau$ increases with $\Omega^d$ and with the droplet radius. Similarly, we plot on Fig. 3 (d) the torque by unit volume $\tau_v = \tau/(\frac{4}{3}\pi R^3)$ which appears to collapse all the data onto a unique curve. This curve corresponds to the operating curve of the droplet motor, analogous to the ones of macroscopic synchronous motors, pointing out a direct link between the core rotation and the flow generation outside the droplet. This collapse means that $\tau_v$ and $\Omega^d$ have a similar dependency with respect to the parameters of our experiments, meaning the droplets radii and the magnetic field intensities. Within experimental uncertainties, the inversion symmetry of this curve is clearly visible with possibly a small bias towards CCW rotation. This implies that the symmetry breaking mechanism acts mainly at the

selection of the direction of rotation, but only weakly on the operation. The non-linear shape of this operating curve shows that the motor is less efficient at low $\Omega^d$ (typically for $\Omega^d < 0.05$ rad s$^{-1}$) than at high $\Omega^d$. At low $\Omega^d$, corresponding to a low magnetic field, the motor is less efficient because the inner vortex structure is not well established. On top of this, Brownian motion on the outer tracers makes the torque measurements quite noisy.

**Mechanism of torque generation**. A global circulation, resulting from a net torque acting on the oil phase and produced by torque-free and force-free swimmers, can only be sourced in the misalignment dynamics of the magnetic moments of the MTB with the external magnetic field. In a quiescent fluid, bacteria align with the field in a time $t_B = \xi_r/(mB)$, where $\xi_r = \pi\eta_w\ell^3/[3\ln(2\ell/a)]$ is the rotational friction coefficient, $l = 3 - 4$ μm is the body length of one bacterium, $a = 1$ μm its width and $\eta_w = 1 \times 10^{-3}$ Pa.s is the dynamic viscosity. With these typical values, we find that $t_B \approx 0.1$ s for $B = 2$ mT. When bacteria swim close to the droplet interface, they are forced to turn in order to align along the interface before swimming parallel to the boundary (this alignment has been clearly observed for dilute suspensions, see Supplementary Note 2 and Supplementary Movie 8). The time needed for one bacterium to orient along the boundary is $t_T = \ell/V_0 \approx 0.1$ s, which is of the same order of magnitude as $t_B$. Hence, the droplet boundary makes the bacteria at the interface misaligned with the magnetic field, which leads to the generation of torque in the droplet. This misalignment depends on the position of the bacteria in the droplet and is

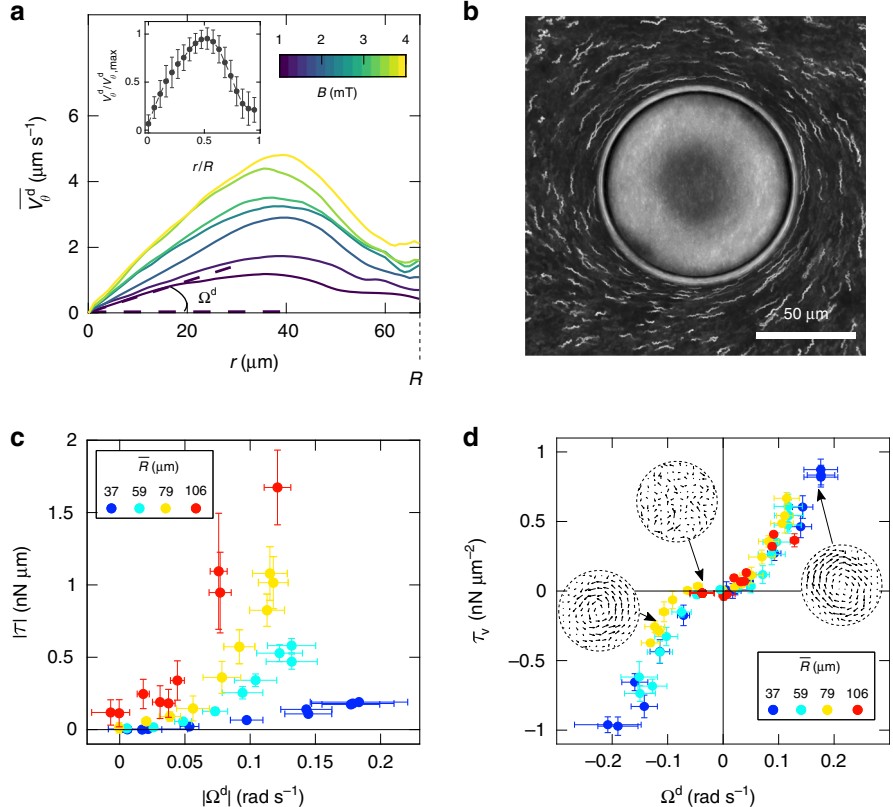

**Fig. 3** Mechanical characterization of the rotary motor. **a** Mean orthoradial velocity profile $\overline{V_\theta^d}(r)$ for one droplet of radius $R = 67\,\mu m$ and for different magnetic field magnitudes $B$ (colors, from bottom to top $B = 1$, 1.4, 2, 2.4, 3, 3.4, 4 mT). Close to the droplet core ($r = 0$), the suspension rotates like a solid with a characteristic rotational velocity $\Omega^d$ which increases with $B$. The error bars are the standard errors. **b** Superposition of phase-contrast images (350 images corresponding to a 14 s movie) showing the circular rotation of the outer tracers for an inner rotational velocity $\Omega^d = 0.13$ rad s$^{-1}$ measured at $B = 4$ mT. **c** The torque $\tau$, acting on the oil and produced by the droplets, is extracted from the tracers orthoradial velocities (see (**b**)). We measure $\tau(B, R)$ for different droplets radii $R$ and magnetic field $B$ with respect to the core rotation velocity $\Omega^d(B, R)$ (average data for 10 droplets of similar radii, $\overline{R}$ is indicated by colors and is given with a $\pm 15\,\mu m$ standard deviation). The error bars are $\sigma/\sqrt{N}$ where $\sigma$ is the standard deviation and $N = 10$ (number of droplets used for the average). **d** Torque by unit volume $\tau_v = \tau/(\frac{4}{3}\pi R^3)$ as a function of $\Omega^d$ for the same data set. The average data for different $\overline{R}$ collapse on the operating curve of the rotary motor. The velocity maps are the ones of the droplet displayed on Fig. 2**d**-**f** and placed at the corresponding operating points. The error bars are the standard errors

expected to be the strongest at the north and south poles of the droplet, where the interface is perpendicular to the magnetic field direction. To precise the kinematics involved inside the droplet, we can take the example of a droplet rotating CW without loss of generality. In this case, due to the advection by the central vortex core and to the swimming propulsion along the magnetic field direction, NS (resp. SS) bacteria reach the droplet boundary and align with it at the right (resp. left) of the NP (resp. SP) in the $x$-$y$ plane. The magnetic torque on those misaligned bacteria (both at the NP and SP) points in the CW direction, thus reinforcing the circulation. As NS (resp. SS) bacteria swim along the droplet boundary, their misalignment with **B** increases while getting close to the NP (resp. SP). Then, trespassing the NP (resp. SP), the situation becomes unstable because the magnetic torque is too large to be compensated by the boundary alignment and also, because they will meet a flow of bacteria transported by the global rotation. Then, these bacteria changing orientation will leave the counter-rotating droplet boundary to be advected by the vortical flow. This orientation flip will, most probably, cause a strong release of magnetic torque in the fluid.

This picture is consistent with the previously mentioned counter-rotating flows inside and outside the droplet. From this time-scale analysis, we can therefore infer that the net torque is produced by the bacteria misaligned with the magnetic field at the droplet boundary, which points on a surface effect. More precisely, we are able to state that:

$$\tau = n\mathcal{V}mB, \tag{2}$$

where $\mathcal{V}$ is the volume of bacteria contributing effectively to the torque. To account for surface effects, we expect $\mathcal{V} \sim \lambda R^2$, where $\lambda$ is a typical length independent of $R$. Indeed, by computing $\lambda = \tau/(nR^2mB)$ (see Fig. 4a), we bring evidence of a characteristic length $\lambda = 8 \pm 2\,\mu m$ that does not depend neither on the magnetic field intensity nor on the droplet radius, collapsing all the data we collected. $\lambda$ is of the order of a bacterium size and can then be related to a microscopic scale, consistent with no dependence on neither $R$ nor $B$. Then, the picture emerging from our scaling analysis is that of one core rotation and two counter-rotating regions made of self-assembling bacteria at the poles and yielding a net torque to the oil (see Fig. 4 (b)). The active torque generated in the droplet $\tau$ would then correspond to an engine mechanical power estimated through the flow generated in the oil phase as: $\mathcal{P} \approx \tau^2/(\eta_{oil}R^3)$. For a typical torque of $\tau = 1$ nN $\mu m$, oil viscosity $\eta_{oil} \approx 3 \times 10^{-3}$ Pa s and for a radius $R = 80\,\mu m$, the estimation yields $\mathcal{P} \approx 10^{-16}$ W.

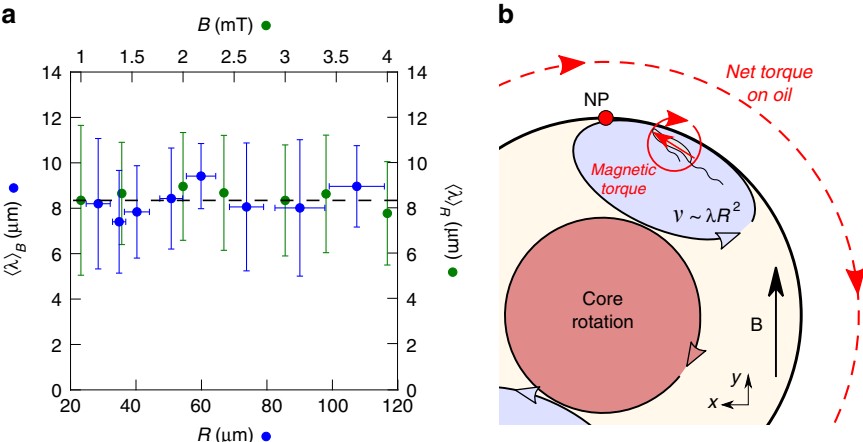

**Fig. 4** Test of the scaling relation: $\tau = nmB\lambda(R,B)R^2$, where $\tau$ is the generated torque, $n \sim 10^{17}$ bact m$^{-3}$ is the bacteria density, $m \sim 10^{-16}$ J T$^{-1}$ is the magnetic moment of a single bacterium, $B$ is the magnetic field intensity and $\lambda(R,B)$ is a typical length inherent to the torque generation. **a** $\langle\lambda\rangle_B$ (resp. $\langle\lambda\rangle_R$) is the average value of $\lambda$ with respect to $B$ (resp. $R$). We only included the data corresponding to $\tau_v > 0.1$ nN µm$^{-2}$, for which the torque is strong enough to be measured out from experimental noise. The error bars are the standard deviations of the average data. This graph shows that $\lambda = 8 \pm 2$ µm is an intrinsic length of the system which does not depend on $R$ nor $B$. The error bars are the standard errors. **b** Qualitative interpretation of the rotary motor self-organization. The volume of the recirculating bacteria contributing to the torque is dimensionally $\mathcal{V} \sim \lambda R^2$. The picture is similar close to the south pole of the droplet

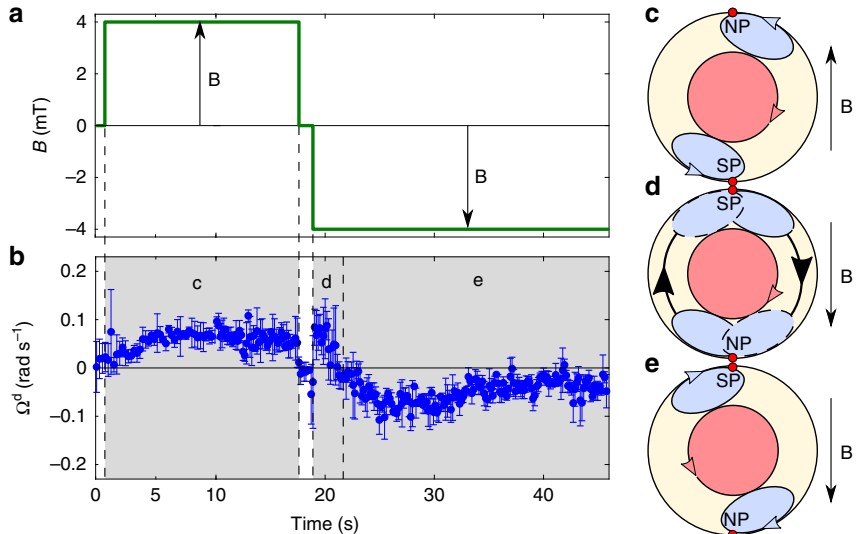

**Fig. 5** Vortex emergence and rotation reversal by magnetic field inversion. **a** Magnetic field amplitude $B$ as a function of time. **b** Response in the rotational velocity of the droplet core $\Omega^d$ under the applied magnetic field. The error bars are the standard errors. **c**, **d**, **e** The stages corresponding to the particular self-assemblies representations on the sketches on the right of the figure. The coral region corresponds to the central core rotation while the blue zones represent the recirculating bacteria. The bacterial suspension is dense ($n \sim 10^{17}$ bact m$^{-3}$). For each time step $t$, $\Omega^d(t)$ is computed from instantaneous PIV maps as in Fig. 3a. From the moment when the magnetic field is set on, $\Omega^d$ reaches a stationary value within a few seconds ($\sim 4$ s). When reversing quickly the magnetic field direction while the suspension rotates CW, the suspension continues rotating CW at the short times after reversal before reversing completely its rotational direction to CCW. Then, $\Omega^d$ reaches a stable negative value $\sim 10$ s after the magnetic field switch

**Vortex reversal.** From our understanding of the bacteria self-assembly inside the droplet, we can provide an explanation of the rotation reversal actuated by the magnetic field switch mentioned above. When the magnetic field is rapidly reversed, the MTB that are accumulated at the right of the NP and at the left of the SP can suddenly rotate to align along the new magnetic field, without any restriction from the droplet boundary. After this, they cross the droplet roughly along the $y$-direction, producing a transient convective circulation with positive values of $\Omega^d$ (corresponding to CW direction) measured just after the magnetic field reversal (see Fig. 5 and

Supplementary Movie 9). Then, they accumulate near the new NP and SP, which are located on opposite positions to the original ones. In this process, the MTB that were accumulated CW of their respective poles are now CCW of the new poles and, hence, generate an opposite torque provoking the inversion of the motor (see the full quantitative report of this dynamics on Fig. 5). For this mechanism to be efficient and overcome the natural tendency to generate a CW motor, the time duration between the application of the two opposite magnetic fields must be shorter than the thermal reorientation time (of the order of the inverse rotational diffusion constant,

$D_r = 1/80 \text{ s}^{-1}$), as it is indeed the case on Fig. 5. Otherwise, the MTB have time to reorient isotropically and leave the accumulation zones.

## Discussion

Recently, considerable efforts have been undertaken to harness the microscopic activity of living or synthetic agents like bacteria[21,44], eukaryotic cells[45], Janus colloids[9] or micro-robots[46,47], in order to extract macroscopic work from microscopic mechanical structures. Here, we show a remarkable example of living biological entities self-assembling into a rotary motor actuated by a controlled, external aligning field. In this process of self-organization, confinement plays an essential role. It would probably be fruitful to exploit it by an adapted design of a micro-fluidic environment. To obtain such a complex organization, a first key-point is the ability for MTB to accumulate in some areas (here being close to the droplet poles) under a uniform magnetic field. This accumulation of active swimmers, obviously limited by crowding, results in an instability triggering a coordinated motion at the scale of the droplet. The second crucial point relies on the original properties of this active magnetic fluid. Any swimming kinematics inducing a change in the magnetic moment orientation of the bacteria with respect to the applied magnetic field, is bound to produce torque on the surrounding fluid. Interestingly, this situation is reversed compared with the standard synchronous motor, exemplified by the magnetic stirrer of chemical labs, which follows the rotation of a rotating magnetic field. In our case, a swimming bacterium carrying a magnetic moment is rotated and creates a torque due to the confinement in a droplet, while the magnetic field direction remains fixed. Even though the physical origin of the self-organization process is not completely elucidated yet, we expect a similar behavior for other types of autonomous swimmers, confined and orientable by any external field (electric field, light, ...), providing new routes of theoretical and experimental investigations.

## Methods

**Bacteria growth protocol**. We used MTB from the MSR-1 *Magnetospirillum gryphiswaldense* strain. MTB are grown in a Flask Standard Medium (FSM) in the absence of external magnetic field, though the Earth magnetic field is still present. This medium was beforehand bubbled with a gas containing 2% $O_2$ and 98% $N_2$ and sealed inside Hungate tubes of 12 mL. We use inoculation volume of 300 µL to start bacteria growing in a tube. In such conditions, we got roughly 50% of NS and 50% of SS MTB in the suspension, consistent with a standard growth protocol of MSR-1[41]. Bacteria used for the experiments shown here are harvested at the end of the growing sigmoid in order to work with the most motile swimmers (bacteria concentration corresponding to an optical density OD = $0.12 \pm 0.02$ measured at a wavelength of 600 nm).

**Emulsion preparation and setup control**. The MTB are initially in a flask standard medium and eventually concentrated by centrifugation. Then, an emulsion is prepared by agitation in the presence of hexadecane oil (*ReagentPlus*, SigmaAldrich) containing Span80 (2%-weight concentration) as surfactant to stabilize the emulsion. We prepare samples at estimated bacterial density from $10^{14}$ to $10^{17}$ bact m$^{-3}$ (volume fractions of 0.01% and 10%, using a bacterial body volume of $\mathcal{V}_b = 3 \text{ µm}^3$). This bacterial number density is estimated for a given OD (optical density), low enough to count the bacteria using phase-contrast images, hence giving a conversion between OD and bacteria concentration. A volume of 65 µL of the emulsion is then deposited in a chamber composed of a double-sided tape adhered to a microscope slide and sealed with a glass cover slip on top (see Supplementary Note 3). This system creates a closed pool of area 1.5 cm × 1.6 cm and height $H = 270$ µm. The emulsion is visualized inside the pool using an inverted microscope adapted to receive Helmholtz coils, which produce a uniform horizontal magnetic field **B** (see Fig. 1a). That is, the visualization plane $(x, y)$ is parallel to **B**. The intensity of the magnetic field $B = |\mathbf{B}|$ is controlled electronically, from 0 to 4 mT with a precision of 0.1 mT. Droplets of diameters smaller than 270 µm sediment at the bottom surface of the chamber due to the low density of hexadecane oil. For a given emulsion preparation, several droplets are visualized in their equatorial plane with respect to the vertical direction (see Fig. 1b). For this report, we use mostly a ×40 phase-contrast objective (Zeiss A-Plan Ph2 Var2, mounted on a Zeiss AXIO Observer microscope) which allows full visualization of the droplets. For all the experiments, we only observe droplets sufficiently separated from each other (typically distant of, at least, one droplet diameter) to avoid any coupling effects between droplets. Experiments are always performed within 30 min after centrifugation for the largest bacteria concentration (longer observations have shown a decrease of bacteria motility after this time). For flow visualization outside the droplets, we used 1.1 µm melamine resin beads suspended in hexadecane oil.

**Data acquisition and analysis**. Phase-contrast images are recorded using a Hamamatsu ORCA Flash4 camera equipped with a CCD sensor of 2048 × 2048 pixels. For movies, a frame rate of 25 Hz is chosen to capture the full dynamics inside and outside the droplets. To prepare PIV analysis, we post-process raw images from experiments by subtracting the average-intensity image of a stack to all the images of the stack. This allows us to get rid of the intensity gradients (which could lead to discrepancies in the flow measurements) inherent to both phase-contrast microscopy and the spherical shape of the droplets. It also provides better accuracy on the velocity map close to the droplet interface. We choose interrogation window size to be equal to 32 × 32 pixels (corresponding to 5 × 5 µm) and an overlap between windows equal to the half of the size of a window. A standard FFT cross-correlation algorithm is used to compute the PIV velocity field using the Matlab PIVlab facilities. We compare successive images separated by 1/25 s. In order to compute the average ortho-radial velocity profile $\overline{V_\theta^d}(r)$, we average the velocity field on a movie of typically 350 images (14 s). For tracking, we used the TrackMate plugin of Fiji (extension of ImageJ). To smooth thermal noise between two successive tracking points in time, we average the velocity of each tracked particle on two successive images. To get the orthoradial velocity field, we compute the circulation $\mathcal{C}(r, t) = \int_0^{2\pi} r V_\theta^{\text{oil}}(r, \theta) \, d\theta$ of the experimental velocity field on circles of various radii $r$ centered on the droplet center at time $t$. Then, this circulation is averaged in time on the duration of the movie (typically 14 s) to get the mean circulation $\overline{\mathcal{C}}(r) = \langle C(r, t) \rangle_t$. Then, we obtain the average orthoradial velocity field $\overline{V_\theta^{\text{oil}}}(r) = \overline{C}(r)/2\pi r$, which is the net velocity of the outer flow. The advantages of this method are both to increase accuracy by smoothing Brownian motion of the tracers and to give a reliable estimate of the net torque applied by the droplet on the oil (counter-flows, opposite to the main recirculation flow, are taken into account in $\overline{V_\theta^{\text{oil}}}(r)$).

## Data availability

The authors declare that the data supporting the findings of this study are available within the paper and its supplementary information files.

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

## Acknowledgments
The authors thank Xavier Benoit-Gonin for his creative technical help on the set-up and Thierry Darnige for his determinant assistance on the numerical interfaces. E.C. is grateful to Dirk Schüler for enlightening scientific discussions on MTB and on providing the MSR-1 strain. The authors acknowledge the support of the ANR-2015 BacFlow under Grant No. ANR-15-CE30-0013, Franco-Chilean EcosSud Collaborative Program C16E03, Fondecyt Grants No. 1180791 and 1170411, and Millenium Nucleus Physics of Active Matter of the Millenium Scientific Initiative of the Ministry of Economy, Development and Tourism (Chile).

## Author contributions
B.V. and G.R. did the experiments. B.V., G.R., M.L.C., C.D., R.S. and E.C. participated in the scientific discussion and the writing of the article.

## Competing interests
The authors declare no competing interests.
