## [Peer Review File · Nature Communications]

Reviewers' comments:

Reviewer #1 (Remarks to the Author):

This paper reports on an interesting finding in the collective behaviour of magnetotactic bacteria exposed to an external magnetic field. A vortex formation is observed which can be used as an efficient motor. The physics is explained by a simple model of a rotating sphere. The topic is timely and exciting. The paper is clearly written and reports novel results. I recommend in principle publication in Nature Communications.

Before publication the author should address the following minor points:

1) In the last sentence of the abstract the authors state that they quantitatively extract the mechanical energy from this motor. It would be helpful if an actual number could be presented here.

2) The following paper could be cited as it is related:
Andrey Sokolov, Igor S. Aranson. Rapid expulsion of microswimmers by a vortical flow. Nature Communications, 2016; 7: 11114.

3) There is a typo in Ref 45 "Pöschel".

Reviewer #2 (Remarks to the Author):

Rotary motor self-assembly in a drop: putting magneto-tactic bacterial to work

The paper describes the use of magnetotactic bacteria confined in a water-in-oil drop to self-assemble into a rotary motor. The primary function of the motor is to exert a torque on the oil phase leading to circular rotation of the oil lying just beyond the outer droplet interface. The collective motion of the bacteria within the droplet, which is reversible by inverting the external magnetic field, is studied as a function of bacterial concentration, field strength and droplet radius. The experimental techniques are good and a simple hydrodynamic model is provided to describe the phenomenon. The presentation is generally clear and well-written.

The authors should address the following:

1. Given the high symmetry of the system being studied, the preferential assembly of a CW motor is intriguing. The authors however do not dwell much (beyond speculating on a possible link to the chirality of the cells) on the mechanism behind this preferential assembly. What symmetry breaking effects could lead to this preferential rotation? Does the helical form of the flagella play a role?
2. The observations (and the proposed mechanism) depend on the presence of roughly equal populations of NS and SS bacteria in the droplet that lead to accumulations at the diametrically opposite NP and SP ends. In general the NS and SS populations of MTB need not be approximately equal. The authors should discuss the expected behavior on the collective core motion and oil rotations if the NS and SS populations were unequal and predominantly of one type; what would occur if there was only a NS or SS population?
3. In Fig. 2b it is not clear why cells appear denser to the left of NP and to right of SP rather than right of NP and left of SP as authors claim in other more denser cases.
4. The droplet core velocity appears to reach a stable negative value after ~20 s of flipping the magnetic field. From the plot (Fig. 5) it appears that this rotation is slower than a typical CW rotation speed. How does this rotation compare to a naturally CCW droplet (without flipping B) of a similar size and cell density. Are these velocity differences consistent with the model?
5. Authors statement that the recirculating regions relocate when the magnetic field is reversed is not substantiated with PIV or by another means. It would be helpful to do so
6. It is claimed that bacteria leave the inner droplet boundary when the magnetic torque becomes too large to be compensated by boundary alignment. It is not clear why the bacteria *leave* the boundary when this occurs.
7. It is not clear whether a single recirculation region near each pole is the cause of the core rotation (under the static B field), or is a result of the core rotation. Should there not be two symmetric recirculation regions on either side of the NP, where one circulation could be suppressed by the core rotation while the other is enhanced? For this reason, I question the validity of the proposed mechanism underlying the core rotation.
8. The core rotation (angular velocity) is stated to *saturate at high field* (see bottom of page 3). Is there an explanation for this leveling off (in the low Reynolds number environment)? How does this rotation velocity saturation, relate with the torque in the oil *increasing linearly with the magnetic field* (Equation 2)?

Minor points:

i) Fig 2 a-c green arrows hard to make out in grey background. Suggestion: yellow arrows

Overall, this is an interesting paper. However, the authors should address the points raised above prior to further consideration.

Reviewer #3 (Remarks to the Author):

Rotary motor self-assembly in a drop: putting magneto- tactic bacteria to work

By Vincenti et al

In this manuscript, the authors show the emergence of coordinated motion in a dense suspension of magnetic bacteria (MTB) confined in spherical droplets and following the application of an external magnetic field. The competition between orientation of the MTB and the external magnetic field allow to exert a torque on the external oil phase and therefore the system constitutes a self-assembled rotary motor.

The result is quite remarkable and the ability to self-assemble a rotor and “push/pull” on an external magnetic field to exert a torque is certainly exciting and inventive. Claims in torque-free/force-free systems are however finicky and even after having spent a lot of time on the manuscript, some of those aspects remain ambiguous or unclear for the referee – in particular, is the rotation of tracers outside of the droplet an unquestionable evidence that the system exerts a torque (and uses the external field to that end). The referee will therefore take advantage of the review process to initiate a scientific discussion with the authors and ask for clarification. The referee does not imply that the claim of the authors that the system exerts a torque is wrong, those are sincere scientific questions.

1. It is the point made by the authors that the rotation of the oil outside of the droplet comes from a net torque coming from the release of the magnetic torque of the bacteria into the fluid following their reorientation with the external magnetic field.

A system is torque-free as long as the angular momentum of the entire system (fluid inside and outside as well as bacteria) is conserved. For a fixed no-slip boundary, that will require a net torque on the boundary. If the surface is free to move, it seems that the rotation of the boundary surface is possible and that a flow could decay outside, possibly displacing tracers, in the absence of any external torque. Could the authors comment on that?

For example, if one considered a virtual boundary (or a boundary between two identical but immiscible fluids, hence hydrodynamical invisible), one would expect a flow decaying outside of the droplet – provided a collective flow inside the droplet, and this in the absence of any “external torque” release in the fluid? In the situation of spiral vortices of bacteria in confined pancakes as in (Wioland et al, PRL, 2013), Wioland et al wrote “*Since the oil viscosity is 10 that of water, the interface acts as a nearly no-slip boundary.*” Do the authors claim that, should (Wioland et al, PRL, 2013) be realized in a fluid of identical viscosity as the inner droplet, there would be no flow outside of the droplet?

The fact that the tracers move outside the droplet does not seem a wrong argument on whether the motion is torque-free or exerts a torque. The scaling of the flow and the relative orientation between the orientation of the MTB in the vortex and the outside flow are essential to emphasize as whether the displacement of tracers outside the droplet originates from a torque-free motion or from a torque. From the current presentation, it seems that the authors claim that the motion of the tracers indicate that there must be a torque. The presented data support this claim (notably the hydrodynamic model of a rotating sphere) but it should be stressed by the authors, which elements can only be explained by a torque.

The difference in behavior between this system and a torque-free system developing a vortex as a result of confinement is essential and should be discussed/clarified to show that indeed, the system constitutes a self-assembled rotor.

2. The authors mention that they do not observe collective motion at high density in their system. Can they comment on that aspect? – in particular in the light of the previous experiments that showed collective behavior of bacteria/active microtubules under confinement? Is the concentration too low or does it originate from the 3D geometry of the spherical droplet? Following, could the emergence of collective behavior following the application of the magnetic field be attributed to the fact that it projects the MTB on a 2D plane?
3. *“Although all symmetric planes containing the magnetic field direction could have been chosen by the bacteria, the vortex is actually rotating in the x-y plane. This might be due to a sedimentation process which yields a stable stratified suspension”*. Could the authors perform an experiment on a tilted microscope to confirm this point?
4. The authors claim that solid core spans one-half of the droplet for all radii but Fig.3a. show only one radius. Could the authors add an inset with normalized radii to support their claim?
5. The caption of Fig.4 states that volume torques below $0.2\text{nM}/\mu\text{m}^2$ are too noisy to be accurately represented. However, they represent more or less 50% of the data point on Fig.3d, which makes no mention of their accuracy. Can the authors comment?
6. *“The non-linear shape of this operating curve shows that the motor is less efficient at low Ω_D (typically for $< 0.05\text{ rad/s}$) than at high Ω_D . This point can be explained by the dominance of the Brownian motion over the advection signal of the passive tracers outside the droplets”*. In the light of the previous comment, do the authors confirm the validity of the low torque regime? Do the authors confirm that they believe that the rotor is less efficient in this regime? It seems that the discussion on the thermal noise of tracer is not directly related to the efficiency of the rotor itself.
7. Given the absence of errors bars on SI-Fig.1b, the corrections of the flow field induced by a near a no-slip wall appear as an overkill. On the other hand, in light of the discussion whether the flow is induced by a moving interface (torque free scenario) or a torque, it would be beneficial to see the flow field in the far field – provided this can be extracted accurately from thermal noise.
8. Overall, it would be strongly beneficial to the manuscript if the arguments made by the authors were supported by in situ observation of the bacteria. For example, it seems that the counter-rotating regions discussed by the authors in the mechanism of torque generation could be seen directly as they are visible on the PIV of Fig.2. How do the extension of the blue regions compare with what would come out of the model? Can the authors image the bacteria reorienting as a result of the magnetic torque?

9. The discussion of the model and timescales is elegant and convincing. (No question here, just a sign of appreciation).
10. The authors make the reasonable prediction that the vortex can be reversed provided a “fast enough” reversal. Could the authors, at least qualitatively, test this prediction?
11. In the conclusion, the authors state “*we expect a similar behaviour for other types of autonomous swimmers, confined and orientable by any external field (electric field, light,...), maybe opening a new branch of theoretical and experimental investigations.*” The authors may want to tone down their claim as the orientation of autonomous swimmers can result of “effective” torques, in spite of being torque-free (see Hagen et al, J. Phys. Condens. Matter, 2015).
12. (Minor comment.) “*However, these swimmers cannot provide any momentum nor a net torque to the fluid because they self-propel at almost zero Reynolds number.*” This sentence is oddly placed in the introduction and should be moved to a more adequate position. Overall, the introduction could be more specific.

REVIEWER #1 (REMARKS TO THE AUTHOR):

This paper reports on an interesting finding in the collective behavior of magnetotactic bacteria exposed to an external magnetic field. A vortex formation is observed which can be used as an efficient motor. The physics is explained by a simple model of a rotating sphere. The topic is timely and exciting. The paper is clearly written and reports novel results. I recommend in principle publication in Nature Communications.

Answer: We thank the referee for his/her appreciation of our work.

Before publication the author should address the following minor points

1) *In the last sentence of the abstract the authors state that they quantitatively extract the mechanical energy from this motor. It would be helpful if an actual number could be presented here.*

Answer: From the torque τ measurements, the motor power can be estimated as $P = \tau \omega^2 / \eta_{oil} / R^3$. For a typical experiment we find a torque $\tau = 1 \text{ nN} \cdot \mu\text{m}$. Using the hexadecane viscosity $\eta_{oil} = 3 \text{ mPa} \cdot \text{s}$ and a typical drop radius $R = 80 \text{ nm}$ yields a motive power around $P = 10^{-16} \text{ W}$. We added in the manuscript an estimation of the magnitude of the motive work.

2) *The following paper could be cited as it is related:*

Andrey Sokolov, Igor S. Aranson. Rapid expulsion of microswimmers by a vortical flow. Nature Communications, 2016; 7: 11114.

Answer: We thank the referee for pointing this reference, we added in the text when we describe the vortical flow (section Results: Vortex flow inside the droplets).

3) *There is a typo in Ref 45 "Pöschel".*

Answer: Thanks. We have corrected this typo.

REVIEWER #2 (REMARKS TO THE AUTHOR):

1. Given the high symmetry of the system being studied, the preferential assembly of a CW motor is intriguing. The authors however do not dwell much (beyond speculating on a possible link to the chirality of the cells) on the mechanism behind this preferential assembly. What symmetry breaking effects could lead to this preferential rotation? Does the helical form of the flagella play a role?

Answer: We agree with the referee that we do not have for the moment a clear idea on the origin of this break of symmetry. The relation of this reproducible and robust fact to the helical body rotation is clearly speculative. We just want to highlight the only potential source of symmetry breaking we could think of but we could not find a convincing hydrodynamic argument to sustain this statement. We just wish to leave it for suggestion that could be useful for future interested readers.

In the new version of the paper, we plot on fig 3d a working curve where the sign of the observed core rotation is explicitly taken into account. Within experimental uncertainties the inversion symmetry of this curve is clearly visible with possibly a small bias towards CCW rotation rate but clearly at the edge of noise. This implies that the symmetry breaking acts mainly at the selection of the direction of rotation but only weakly on the operation. We add a comment in the section "Torque measurements".

2. The observations (and the proposed mechanism) depend on the presence of roughly equal populations of NS and SS bacteria in the droplet that lead to accumulations at the diametrically opposite NP and SP ends. In general the NS and SS populations of MTB need not be approximately equal. The authors should discuss the expected behavior on the collective core motion and oil rotations if the NS and SS populations were unequal and predominantly of one type; what would occur if there was only a NS or SS population?

Answer: The objective of the paper was the analysis of the situation where NS and SS are roughly in similar proportion. Indeed, the situation where NS or SS only are confined in droplets was not systematically studied. However, this is a very interesting question and we started to do experiments on that aspect following the referee's comment. For example, selecting NS or SS population could break the symmetry North/South of the problem and maybe lead to a very different collective behavior as the emergence of a double vortex motion, similar to that observed in some active droplets.

The main challenge of selecting NS populations is to select bacteria that swim persistently, during the duration of the experiment, towards the North Pole of the magnetic field. By selecting NS and SS using a macroscopic magnet, we always recovered, at the scale of the droplets (about 100 μm of radius), some SS individuals, but significantly less numerous than the NS ones. Even with such an unbalanced ratio of NS/SS, we found that the suspension collectively rotates, preferentially CW as we found in our paper with equivalent proportions of NS and SS. So, since we do not observe anything spectacular happening, we just mention this briefly in the new version of the paper.

3. *In Fig. 2b it is not clear why cells appear denser to the left of NP and to right of SP rather than right of NP and left of SP as authors claim in other more denser cases.*

Answer: Figure 2b corresponds to a snapshot of an experiment with a bacterial density below the threshold of global rotation. In this regime, jets are formed, which are very unstable and change position continuously as can be seen in the Supplementary Movie S3. It is only after the global rotation is started, when the symmetry has been broken, that the dense regions locate preferentially at the right of NP and left of SP and correspond roughly to the counter flow regions

In Figs. 2. a-b-c, images are phase-contrast observations of droplets that were post-processed to enhance bacteria motion to improve the efficiency of the PIV analysis that detect motion correlations of spatial textures. However, even in the dense regime, we have the impression based on raw images that some local differences in bacteria density within the droplets can be identified. The difficulty of this visualization (and which makes it non-quantitative) is that the phase-contrast image of a droplet, even without bacteria inside, is strongly inhomogeneous in intensity from the centre to the periphery, which is responsible for the dark central region visible on phase-contrast images (see Fig. 1, left). It is however, possible to compare zones that are equidistant to the droplet centre, as inhomogeneity in light intensity between these zones cannot be accounted for by the geometry of the droplet. In phase-contrast images, due to the refractive index difference between the bacteria and the bacterial medium, bacteria in the objective focal plane appear darker than the medium. Within the droplet, dark regions are visible close to the NP (a little bit shifted on the right) and to the SP (shifted on the left). But finally, after a second look, this is indeed a very weak evidence, which does not necessarily bring something crucial to our message. So we decided to remove this remark in the new version as it can bring some confusion to the reader from what is shown in fig. 2.

4. *The droplet core velocity appears to reach a stable negative value after ~20 s of flipping the magnetic field. From the plot (Fig. 5) it appears that this rotation is slower than a typical CW rotation speed. How does this rotation compare to a naturally CCW droplet (without flipping B) of a similar size and cell density. Are these velocity differences consistent with the model?*

Answer: We thank the referee for this interesting remark. Indeed, on Fig. 5 the inner angular velocity measured in average after flipping (CCW: $0.04 \pm 0.02 \text{ s}^{-1}$) could be seen as lower than the one before flipping (CW: $0.06 \pm 0.01 \text{ s}^{-1}$). However, the experimental uncertainties render any firm conclusion quite hazardous. Note that in response to the magnetic field flipping, there is an overshoot appearing on this figure that could lead to some confusion on the actual magnitude of the effect. Unfortunately, our statistics on flipping events is not large enough to state if, in average, flipping would lead to a slower velocity. We then prefer not to build too much of this issue.

Note in addition that the time scale analysis made in this manuscript and based on the alignment competition between the droplet boundary and the magnetic field direction is general and equally valid to both CW and CCW rotation of the droplet core. From our statistics on the droplet rotation (without magnetic field flipping), it seems that CW rotation speed is the same, in average, than the

CCW rotation speed as shown in the new figure 3d.

5. *Authors statement that the recirculating regions relocate when the magnetic field is reversed is not substantiated with PIV or by another means. It would be helpful to do so*

Answer: We agree with the referee that our statement needs to be substantiated more deeply. This is why we provide the SI with a new video bringing direct evidences for this fact. Supplementary Movie 9 shows the x and y component of the velocity field. After the magnetic field is reversed, two currents form, one at the right hand side, going down (in the new direction of the field) and an opposite one at the left hand side. The current on the right side corresponds to NS bacteria that move following the new north direction and the contrary happen for the current on the left.

We updated the reference and the description respectively in the text and in the SI document.

6. *It is claimed that bacteria leave the inner droplet boundary when the magnetic torque becomes too large to be compensated by boundary alignment. It is not clear why the bacteria leave the boundary when this occurs.*

Answer: We agree that we may need to clarify this point. In the new version we developed the following explanation to justify that counter rotating bacteria trespassing the pole limit will most likely reorient before being convected by the global rotation and this reorientation will be a source of torque to the fluid.

“As NS (resp. SS) bacteria swim along the droplet boundary, their misalignment with \vec{B} increases while getting close to NP (resp. SP). Then, trespassing the NP (resp. SP) limit, the situation will become unstable first because the magnetic torque becomes too large to be compensated by the boundary alignment and also, because they will meet a flow of bacteria transported by the global rotation. Then, these bacteria changing orientation will leave the counter rotating droplet boundary to be advected by the vortical flow. This orientation flip will most probably, cause a strong release of magnetic torque in the fluid.”

7. *It is not clear whether a single recirculation region near each pole is the cause of the core rotation (under the static B field), or is a result of the core rotation. Should there not be two symmetric recirculation regions on either side of the NP, where one circulation could be suppressed by the core rotation while the other is enhanced? For this reason, I question the validity of the proposed mechanism underlying the core rotation.*

Answer: The hypothesis of the manuscript is that the recirculation regions near the poles are the cause of the core rotation because it is in these regions where the magnetic torque is acting. This seems to be corroborated by the scaling found for the torque as a function of all the experimental parameters. The crucial point is that the torque sources essentially involve a volume based on a boundary layer and not a bulk term.

Without this torque, there would be no rotation as it can be seen in Fig. 5, Supplementary Movie S7 and in the new Supplementary Movie 9 when the magnetic field is temporally switched off. The only case when the core rotates uncoupled with the recirculation regions is after the magnetic time reversal, when the accumulation regions relocate. In this transient regime, a convective circulation develops but only originated in the migration of the previously referred regions. We have included a more detailed discussion in the manuscript.

In what concerns the second question, on the possible existence of two recirculating regions for each pole, we have not observed this situation in our experiments. The observations indicate the presence of only one recirculating region for each pole. But we do not clearly see why this absence of two symmetrical structure could be dismissing the simple picture we present in this article

8. *The core rotation (angular velocity) is stated to saturate at high field (see bottom of page 3). Is there an explanation for this leveling off (in the low Reynolds number environment)? How does this rotation velocity saturation, relate with the torque in the oil increasing linearly with the magnetic field (Equation2)?*

Answer: Yes indeed the core rotation seems to saturates at high field and this can be visualized in Fig 3d where we observe a significant increase of the torque whereas the rotation rate seems to level off at +/- 0.2 rad/s. We have no model describing the inner rotation dynamics, however in our mind this saturation does not contradict the linear relation between torque and magnetic field.

Minor points:

i) Fig 2 a-c green arrows hard to make out in grey background. Suggestion: yellow arrows

Answer: We have changed the color of the arrows to increase the contrast.

REVIEWER #3 (REMARKS TO THE AUTHOR):

« In this manuscript, the authors show the emergence of coordinated motion in a dense suspension of magnetic bacteria (MTB) confined in spherical droplets and following the application of an external magnetic field. The competition between orientation of the MTB and the external magnetic field allow to exert a torque on the external oil phase and therefore the system constitutes a self-assembled rotary motor.

The result is quite remarkable and the ability to self-assemble a rotor and “push/pull” on an external magnetic field to exert a torque is certainly exciting and inventive. Claims in torque-free/forcefree systems are however finicky and even after having spent a lot of time on the manuscript, some of those aspects remain ambiguous or unclear for the referee – in particular, is the rotation of tracers outside of the droplet an unquestionable evidence that the system exerts a torque (and uses the external field to that end). The referee will therefore take advantage of the review process to initiate a scientific discussion with the authors and ask for clarification. The referee does not imply that the claim of the authors that the system exerts a torque is wrong, those are sincere scientific questions. »

Answer: We thank the referee for his/her positive appreciation of our work.

1. *« It is the point made by the authors that the rotation of the oil outside of the droplet comes from a net torque coming from the release of the magnetic torque of the bacteria into the fluid following their reorientation with the external magnetic field.*

A system is torque-free as long as the angular momentum of the entire system (fluid inside and outside as well as bacteria) is conserved. For a fixed no-slip boundary, that will require a net torque on the boundary. If the surface is free to move, it seems that the rotation of the boundary surface is possible and that a flow could decay outside, possibly displacing tracers, in the absence of any external torque. Could the authors comment on that?

For example, if one considered a virtual boundary (or a boundary between two identical but immiscible fluids, hence hydrodynamical invisible), one would expect a flow decaying outside of the droplet – provided a collective flow inside the droplet, and this in the absence of any “external torque” release in the fluid? In the situation of spiral vortices of bacteria in confined pancakes as in (Wioland et al, PRL, 2013), Wioland et al wrote “Since the oil viscosity is 10 that of water, the interface acts as a nearly no-slip boundary.” Do the authors claim that, should (Wioland et al, PRL, 2013) be realized in a fluid of identical viscosity as the inner droplet, there would be no flow outside of the droplet?

The fact that the tracers move outside the droplet does not seem a wrong argument on whether the motion is torque-free or exerts a torque. The scaling of the flow and the relative orientation between the orientation of the MTB in the vortex and the outside flow are essential to emphasize as whether the displacement of tracers outside the droplet originates from a torque-free motion or from a torque. From the current presentation, it seems that the authors claim that the motion of the tracers indicate that there must be a torque. The presented data support this claim (notably the

hydrodynamic model of a rotating sphere) but it should be stressed by the authors, which elements can only be explained by a torque.

The difference in behavior between this system and a torque-free system developing a vortex as a result of confinement is essential and should be discussed/clarified to show that indeed, the system constitutes a self-assembled rotor. »

Answer: This is a pertinent remark and we agree with the referee that we need to explain better the rationale behind our procedure. We elaborated an answer which, we hope, will improve the clarity of the motivations for our method to extract the external global torques.

The referee is right in principle: the existence of a non-zero circulation is not a mathematical proof that a net torque is exerted on the fluid. For example, a single swimmer inside the droplet will indeed create, in general, a non-zero circulation (see *infra*). It is then clear that for a collection of non-magnetic swimmers inside the droplet, this should be the same.

However, the character of the velocity field and the circulation produced by zero-torque sources (torque-free swimmers) is completely different to those when there are torques acting on the fluid. As we show below, there is a major difference: for a zero-torque source one should observe when going in the vertical direction, a change of rotation direction characterized by a change of sign of the circulation, which we do not observe.

First, let us start with a simple model for the flow circulation around a single swimmer producing a stresslet whose velocity field is: $\mathbf{u}(\mathbf{r}) = (3\cos(\theta)^2 - 1)\mathbf{r}/r^3$ (the dipolar strength was dimensionalized appropriately without loss of generality). The circulation of this velocity field on a circle chosen randomly in space is in general non-zero (see *fig1*). Interestingly, the condition of zero torque on the external fluid implies that the rotation direction should be inverted at some point when going in the vertical direction. This is exemplified on *fig 1* where on the right panel, the circulation on the circles pertaining to a sphere of radius 1.2 are computed. The integral of this circulation is zero since no global torque is exerted on this sphere. We have added a proof on this in the Supplementary Information.

Now we complicate a little bit the model. We still consider one phase (no oil) but the swimmers distributed randomly in space inside a sphere of radius 1 and organized such that their swimming direction performs a global rotation inside the droplet. Each of the force dipoles is oriented at 20° with respect to the local tangent of the circles (see fig. 2). As for a single swimmer, the circulation integrates to zero and displays a change of sign stemming from a change in rotation at some vertical position. In the experiment we do not observe anything like this (see Supplementary Movies 4, 5, and 6). For example, in fig 3, we show two horizontal slices in positions below and above the equatorial plane and displaying a similar rotation.

Fig.2 Left panel - Sphere of radius 1 filled with 4031 swimmers placed in several planes of constant z and along concentric circles. Each of the force dipole is oriented at 20° with respect to the local tangent of the circles. Other angles can be chosen and the phenomenology remains identical. The idea behind the toy model is to recreate a situation of rotating micro-swimmers coherently in the droplet. Right panel - Circulation of the flow field created by all the swimmers on circles of radius 1.2 enclosing the sphere. For this calculation, The circulation is non zero along z but its integral is zero

Fig.3 Global rotation of the oil in planes parallel to the droplet equatorial plane. The maximal intensities of a 15 s movie (25 [fps]) have been summed up to obtain these two images, enhancing the tracers trajectory in oil. The white bar length is $20\ \mu\text{m}$. The droplet is identical for (a) and (b), its radius is 55. The motion of oil is indicated by the arrow (CW) and is in the same direction as the one at the equator. (a). $28\ \mu\text{m}$ below the equator. (b). $40\ \mu\text{m}$ above the equator.

A second aspect that is different for zero-free source is that the produced flow is mainly radial (Eq. 4 of the Supplementary Information), while we observe principally a tangential velocity field.

In summary, although it is possible that zero-torque sources can produce a finite circulation, the character of the velocity field and the circulation produced in this case is completely different to what we experimentally observe. Indeed, the experimental results are consistent with the presence of torque sources in the fluid.

In the Supplementary Information and in Section “Flow in the oily phase” we explain in detail these arguments.

2. *« The authors mention that they do not observe collective motion at high density in their system. Can they comment on that aspect? – in particular in the light of the previous experiments that showed collective behavior of bacteria/active microtubules under confinement? Is the concentration too low or does it originate from the 3D geometry of the spherical droplet? Following, could the emergence of collective behavior following the application of the magnetic field be attributed to the fact that it projects the MTB on a 2D plane? »*

Answer: Thanks for this very interesting question. The mention of the absence of collective motion is when the magnetic field is turned off. In this paper, we just focus on the characteristics of the collective motion when a magnetic field is applied because, indeed, several studies already investigated the collective motion in confined active suspensions without an aligning field.

However, the phrasing was not completely accurate because what we really observed is that, even in absence of magnetic field, some collective motion appears in the form of fluctuating and intermittent vortices, with a typical size much smaller than the droplet diameter. This is in agreement with previous studies. We do not observe, nevertheless, collective motion at the global scale. One difference with the work of Wioland et al, PRL, 2013, is that they use a high volume fraction (0.4), while in our case the highest volume fraction is 0.1. Moreover, we observe, comparing MTB with our experiments with *E. coli*, that collective motion at zero magnetic field are of shorter range, meaning that motion is observed but it is less spatially correlated for MTB than for *E. coli*. This could be due to the behavior of MTB at high density (less motile than *E. coli* at similar volumetric fractions).

We rephrase the text at the beginning of section “Results: Vortex flow inside the droplets” and compare our results with that of Wioland et al.

What is true is that the alignment of the bacteria with the magnetic field breaks the spherical symmetry of the system. From our observations, the magnetic field plays a crucial role in the emergence of the collective effect. First because it focuses bacteria at the poles of the droplet which creates hydrodynamic « jets » propelling the fluid at the poles. These hydrodynamic jets are very important to initiate and maintain the collective rotation of the bacteria in the droplet. The focusing of the bacteria at the poles is however not enough to explain why a vortex is observed stable in the observation plane. That is why we claim that this comes from the stratification of the suspension by gravity, MTB being denser than their medium, which was verified by microscopic scans in dilute suspensions.

3. « *“Although all symmetric planes containing the magnetic field direction could have been chosen by the bacteria, the vortex is actually rotating in the x-y plane. This might be due to a sedimentation process which yields a stable stratified suspension”. Could the authors perform an experiment on a tilted microscope to confirm this point? »*

Answer: We thank the referee for this suggestion. However, performing this experiment was quite difficult and a little bit disappointing. Indeed, the suspension of bacteria is much denser than the hexadecane oil. When tilting the microscope, we observed a sliding of the droplets due to gravity, which then stick to the edge of the sample pool. This made impossible an observation of a single isolated droplet.

4. « *The authors claim that solid core spans one-half of the droplet for all radii but Fig.3a. show only one radius. Could the authors add an inset with normalized radii to support their claim? »*

Answer: This was added to the manuscript where an average velocity plot over 40 droplets and several values of the magnetic field (about 300 orthoradial velocity profiles) was introduced in an inset on Fig. 3a.

5. « *The caption of Fig.4 states that volume torques below $0.2\text{nN}/\mu\text{m}^2$ are too noisy to be accurately represented. However, they represent more or less 50% of the data point on Fig.3d, which makes no mention of their accuracy. Can the authors comment? »*

Answer: There is a typo in the limit chosen on Fig.4. It is $0.1\text{ nN}/\mu\text{m}^2$ (this has been corrected). This limit of torques was chosen considering the results reported on the Fig. 3d graph. On this graph, one can notice that, below $0.1\text{ nN}/\mu\text{m}^2$ (or below an inner rotation of $0.05\text{ rad}\cdot\text{s}^{-1}$), the rotation of the suspension induces a low torque. These low torque values are not very accurate considering the fluctuations in the circulations mentioned above in this report. This regime, which occurs for an inner rotation of the suspension between and -0.05 s^{-1} and 0.05 s^{-1} , corresponds to a low efficiency motor regime where the central vortex is not well established (this was verified by PIV). As the simple model we propose is based on a particular self-assembly of the suspension, we decided to consider only data beyond the torque threshold of $0.1\text{nN}/\mu\text{m}^2$ in order to compute the effective length scale λ .

6. « *“The non-linear shape of this operating curve shows that the motor is less efficient at low Ω_D (typically for $< 0.05\text{ rad/s}$) than at high Ω_D . This point can be explained by the dominance of the Brownian motion over the advection signal of the passive tracers outside the droplets”. In the light of the previous comment, do the authors confirm the validity of the low torque regime? Do the authors confirm that they believe that the rotor is less efficient in this regime? It seems that the discussion on the thermal noise of tracer is not directly related to the efficiency of the rotor itself. »*

Answer: This is completely correct, meaning that the discussion on thermal noise is not directly related to the efficiency of the motor at low Ω_d . In fact, we wanted to stress that, at low

Ω_d , the motor is less efficient because the inner vortex structure is not well established for these so low Ω_d , which correspond to low magnetic field intensities. On top of this low efficiency, the Brownian motion of the outer tracers makes the torque measurements noisy. We restructured this sentence to make it less ambiguous.

7. *«Given the absence of errors bars on SI-Fig.1b, the corrections of the flow field induced by a near a no-slip wall appear as an overkill. On the other hand, in light of the discussion whether the flow is induced by a moving interface (torque free scenario) or a torque, it would be beneficial to see the flow field in the far field – provided this can be extracted accurately from thermal noise. »*

Answer: On SI-Fig.1b, no error bars were included for clarity reasons, which are now included in the resubmitted version of the manuscript. The figure shows that the fit is adequate and not as an overkill. It is true that it would be desirable to measure the velocity field in the far field; however, this is not possible because the weak signal is masked by the Brownian noise, the interactions with other droplets and, finally, the presence also of the top wall. It appears that, for all the data we obtained, the fit considering the bottom boundary is always better than the fit for a rotating sphere in the bulk. Then, the correction brought to a no-slip wall seems to reflect a real physical effect.

8. *«Overall, it would be strongly beneficial to the manuscript if the arguments made by the authors were supported by in situ observation of the bacteria. For example, it seems that the counterrotating regions discussed by the authors in the mechanism of torque generation could be seen directly as they are visible on the PIV of Fig.2. How does the extension of the blue regions compare with what would come out of the model? Can the authors image the bacteria reorienting as a result of the magnetic torque? »*

Answer: Indeed, it would be beneficial to have a direct observation of the bacteria. However, the phenomenon we report takes place in the dense regime, where it is not possible to observe them individually and follow their motion. We add a short note about this in the text (section “Results.Vortex flow inside the droplets”).

Determining the extension of the counter-rotating regions in the observation plane is indeed possible since we have the PIV fields. We had already thought about measuring the extension of the counter-rotating region and how the measured torque could possibly scale with it. However note we do not have access to their extension in the z direction. Hence, its volumetric extension can only be guessed. Anyhow, we tried to correlate the area with the torque. A weak signature in favor of a linear correlation is obtained, but we found the results to noisy to be able to stand on a strong quantitative statement. In consequence, we preferred not to present these results.

9. *«The discussion of the model and timescales is elegant and convincing. (No question here, just a sign of appreciation).»*

10. « *The authors make the reasonable prediction that the vortex can be reversed provided a “fast enough” reversal. Could the authors, at least qualitatively, test this prediction?* »

Answer: This statement is not a really prediction but the results of measurement and is consistent with the simple picture we develop in the manuscript. We rephrased this sentence to clarify our point and compare the reversal time to the Brownian reorientation time. Also we provide in SI a visualization of the velocity orientation field under reversal that sustains this statement.

11. « *In the conclusion, the authors state “we expect a similar behavior for other types of autonomous swimmers, confined and orientable by any external field (electric field, light,...), maybe opening a new branch of theoretical and experimental investigations.” The authors may want to tone down their claim as the orientation of autonomous swimmers can result of “effective” torques, in spite of being torque-free (see Hagen et al, J. Phys. Condens. Matter, 2015).* »

Answer: According to your suggestion, we have toned down the claim.

In that respect considering the article by Hagen et al., however interesting it is, there is a substantial difference with our work. Indeed, they introduce effective torques acting on the particles, while in our experiment we describe the emergence of real torques.

12. « (Minor comment.) “However, these swimmers cannot provide any momentum nor a net torque to the fluid because they self-propel at almost zero Reynolds number.” This sentence is oddly placed in the introduction and should be moved to a more adequate position. Overall, the introduction could be more specific.

Answer: We have moved this sentence, with some modifications, to the end of the introduction.

REVIEWERS' COMMENTS:

Reviewer #2 (Remarks to the Author):

The authors have adequately addressed the questions raised in my first report. I recommend publication.

Reviewer #3 (Remarks to the Author):

The referee thanks the authors for the thorough response. The circulation argument is compelling and convincing to address the question raised by the referee on the rotation induced by torque-free systems.

The referee now comfortably recommends publication in Nature Communications.

However, the sections added to address the comments/questions of the referees (in red) could be more polished. They present numerous typos and the English appears more colloquial (or loose) than in the rest of the manuscript. The authors should improve this point and make the additions as clear and smooth as the rest of the manuscript.